# Chemicals orchestrate reprogramming with hierarchical activation of master transcription factors primed by endogenous *Sox17* activation

Zhenghao Yang[1,9], Xiaochan Xu[2,9], Chan Gu[3,9], Jun Li[2], Qihong Wu[3], Can Ye[4], Alexander Valentin Nielsen[5], Lichao Mao[1], Junqing Ye[6], Ke Bai[1], Fan Guo[3✉], Chao Tang[2,7,8✉] & Yang Zhao[1,2,4✉]

Mouse somatic cells can be chemically reprogrammed into pluripotent stem cells (CiPSCs) through an intermediate extraembryonic endoderm (XEN)-like state. However, it is elusive how the chemicals orchestrate the cell fate alteration. In this study, we analyze molecular dynamics in chemical reprogramming from fibroblasts to a XEN-like state. We find that *Sox17* is initially activated by the chemical cocktails, and XEN cell fate specialization is subsequently mediated by Sox17 activated expression of other XEN master genes, such as *Sall4* and *Gata4*. Furthermore, this stepwise process is differentially regulated. The core reprogramming chemicals CHIR99021, 616452 and Forskolin are all necessary for *Sox17* activation, while differently required for *Gata4* and *Sall4* expression. The addition of chemical boosters in different phases further improves the generation efficiency of XEN-like cells. Taken together, our work demonstrates that chemical reprogramming is regulated in 3 distinct "prime–specify–transit" phases initiated with endogenous *Sox17* activation, providing a new framework to understand cell fate determination.

[1] State Key Laboratory of Natural and Biomimetic Drugs, MOE Key Laboratory of Cell Proliferation and Differentiation, Beijing Key Laboratory of Cardiometabolic Molecular Medicine, Institute of Molecular Medicine, Peking University, 100871 Beijing, China. [2] Peking-Tsinghua Center for Life Sciences, Peking University, 100871 Beijing, China. [3] Center for Translational Medicine, Ministry of Education Key Laboratory of Birth Defects and Related Diseases of Women and Children, Department of Obstetrics and Gynecology, West China Second Hospital, Sichuan University, 610041 Chengdu, Sichuan, China. [4] BioMed-X center, Peking University, 100871 Beijing, China. [5] The Niels Bohr Institute, University of Copenhagen, Blegdamsvej 17, 2100 Copenhagen, Denmark. [6] Boehringer Ingelheim International GmbH (China), 100027 Beijing, China. [7] Center for Quantitative Biology, Peking University, 100871 Beijing, China. [8] School of Physics, Peking University, 100871 Beijing, China. [9] These authors contributed equally: Zhenghao Yang, Xiaochan Xu, Chan Gu. ✉email: guofan@scu.edu.cn; tangc@pku.edu.cn; yangzhao@pku.edu.cn

Somatic cells can be reprogrammed to become pluripotent by nuclear transfer into oocytes, by delivery of transcription factors or by treatment with a cocktail of chemicals[1–3]. These somatic reprogramming techniques hold great promise in regenerative medicine for providing an unlimited source for functional cells. In comparison with the other two strategies, chemical reprogramming is attractive for future applications due to its non-integrative nature, ease to be standardized and temporally controlled, and lower tumorigenicity[3,4].

In recent years, the understanding of the cell dynamics and the molecular mechanisms of chemical reprogramming has gone deeper and broader. For instance, an extraembryonic endoderm (XEN)-like state bridges the chemical reprogramming towards chemically reprogrammed into pluripotent stem cells (CiPSCs) from different somatic cell types[4,5]. Dynamic early-embryonic-like programs are found critical for the transition of XEN-like state into a pluripotent state[6]. In addition, the chemical reprogramming efficiency has been found greatly improved by additional chemical boosters, such as bromodeoxyuridine, retinoic acid agonists, Dolt1L inhibitors, and glycogen synthase kinase 3 inhibitors, and CiPSC can even be induced with a chemically defined medium[4,7–9].

Furthermore, chemical reprogramming strategies have been extended to inducing direct cell lineage conversion into functional cell types without an intermediate pluripotent state. For instance, neural progenitors[10], functional neurons[11,12], cardiomyocytes[13,14], skeletal muscles[15], brown adipocytes[16,17], astrocytes[18], endoderm progenitor-like cells[19], and photoreceptor-like cells[20] are reported to be induced from fibroblasts with chemicals alone. Besides, endoderm progenitor cells are induced from gut epithelium with pure chemicals[21], and human fetal astrocytes are converted into functional neurons by chemical combinations[22].

Intriguingly, the small molecules essential for XEN induction, CHIR99021 (a GSK3 inhibitor), 616452 (Repsox, an ALK5 inhibitor), and Forskolin (a cAMP agonist) have frequently been used for the direct induction of the many of different cell types noted above. Unlike the master genes used in transgenic reprogramming, which are associated with the target cell type, these chemicals always target signaling pathways that play roles in different cell types and are not associated with any specific cell lineage. Therefore, it is still unclear how the chemical cocktails determine the target cell type, and the molecular dynamics during chemically induced cell fate transition are still elusive[23].

Here, to better understand how chemically induced cell fate alteration and determination are orchestrated, we studied the chemical reprogramming process from fibroblasts to XEN-like cells in terms of the time-course and at the single-cell resolution. We revealed that cell fate transition was primed by endogenously expressed Sox17, which mediated further hierarchical activation of master transcription factors in chemical reprogramming. We further investigated the role of small molecules in various stages throughout the process.

## Results

**Chemically induced *Sox17* expression initiates XEN-like cell fate reprogramming.** To investigate how the chemical cocktail determines XEN-like cell fate during C6FAE-mediated reprogramming (C, CHIR99021; 6, 616452; F, Forskolin; A, AM580; E, EPZ004777) (Fig. 1a), we analyzed the reprogramming process at 10 time points over a course of 20 days with single-cell RNA-sequencing (Fig. 1b). In comparison with the existing dataset[6], our data detected more UMIs and genes, and the expression pattern of XEN and fibroblast master genes in various periods were comparable (Supplementary Fig. 1a–d). Importantly, MEF cells and XEN cells (day 20) merged perfectly with those in the

existing dataset (Supplementary Fig. 1e), indicating the fidelity of our single-cell RNA-seq data.

By principal component analysis (PCA) of single-cell RNA-seq data from days 0 to day 20, we found that cells in the earlier stage were quite close to each other and then separated gradually, and ultimately divided into two branches (Fig. 1c). This bifurcated reprogramming trajectory was also confirmed by pseudo-time analysis (Supplementary Fig. 1f). These two branches mainly consisted of cells from day 12 and day 20 and they belonged to different clusters in which cells were grouped using Louvain clustering with a resolution 0.95[24–26] (Supplementary Fig. 1g). We evaluated cell identities by using the analytical technique based on quadratic programming with 100 genes representing for MEF cell identity and 100 genes representing for XEN cell identity[27]. The left branch had established the major XEN identity without MEF identity (Fig. 1d and Supplementary Fig. 1h), indicating successful reprogramming into XEN-like cells and was further referred to as "the proceeding branch". While, the right branch reserved the major profiles of MEF identity, without the establishment of XEN identity, which was further termed as "the trapped branch". The cells located at the proceeding branch had a remarkable higher expression of the XEN master genes *Sox17*, *Sall4*, and *Gata4*, which were reported to promote the differentiation into XEN cells from mouse embryonic stem cells by forming a self-activation loop[28–31] (Fig. 1e). The trapped branch included cells with a lower expression of *Sall4* and *Gata4* while still retained the high expression of fibroblast master genes, such as *Osr1*, *Prrx1*, and *Twist2* (Fig. 1e). Interestingly, very few cells had low scores of MEF identity before XEN-like cells were induced. This indicates no distinct de-differentiated, or other kinds of intermediate, cells during chemical reprogramming from MEFs to XEN-like cells (model 1–4 in Fig. 1a).

We noticed that the major differences between the proceeding and the trapped branches were the differential expression of XEN and fibroblast master genes. The expression of XEN master transcriptional factors (TFs), especially *Sox17*, were enriched in the proceeding branch (Fig. 1e and Supplementary Fig. 1i). Besides, the order of the activated expression of XEN master TFs was *Sox17*, *Sall4*, *Gata4*, and *Foxa2*, suggesting that *Sox17* was upstream of the other XEN TFs (Supplementary Fig. 1j–l).

In line with the above, we found that *Sox17* knockdown impaired the activation of most of the XEN TFs, *Gata4*, *Sall4*, and *Foxa2*, as well as XEN-like colony formation (Fig. 1f). Meanwhile, *Sox17* overexpression promoted the upregulation of *Sall4, Gata4*, and *Foxa2* (Fig. 1g). Then, we analyzed the co-expression of XEN master gene expression every day throughout the reprogramming process by immunofluorescence. We found that the expression of Sox17 was detected as early as day 4. Sall4, Gata4, and Foxa2-expressing cells were all subpopulations of Sox17-expressing cells that emerged in day 5-8 (Fig. 1h), indicating Sall4, Gata4, and Foxa2 were only activated in Sox17-positive cells.

These findings suggest that chemical-mediated XEN-like cell reprogramming is mediated by the endogenously activated *Sox17* in fibroblasts (Fig. 1i).

**XEN-like cell fate specification and transition with the accumulated master TFs downstream of *Sox17*.** To further investigate how the cell fate reached a XEN-like state after Sox17 activation, we focused on the activation of Gata4, Sall4, and Foxa2. We found that Gata4-positive cells were a subset of Sall4-positive cells and no Gata4-positive/Sall4-negative cells appeared before day 6 by analyzing the co-expression of Gata4 and Sall4 using immunostaining. This suggests Gata4 activation is only in Sall4-expressing cells (Fig. 2a, b). Afterward from day 7 to day 12,

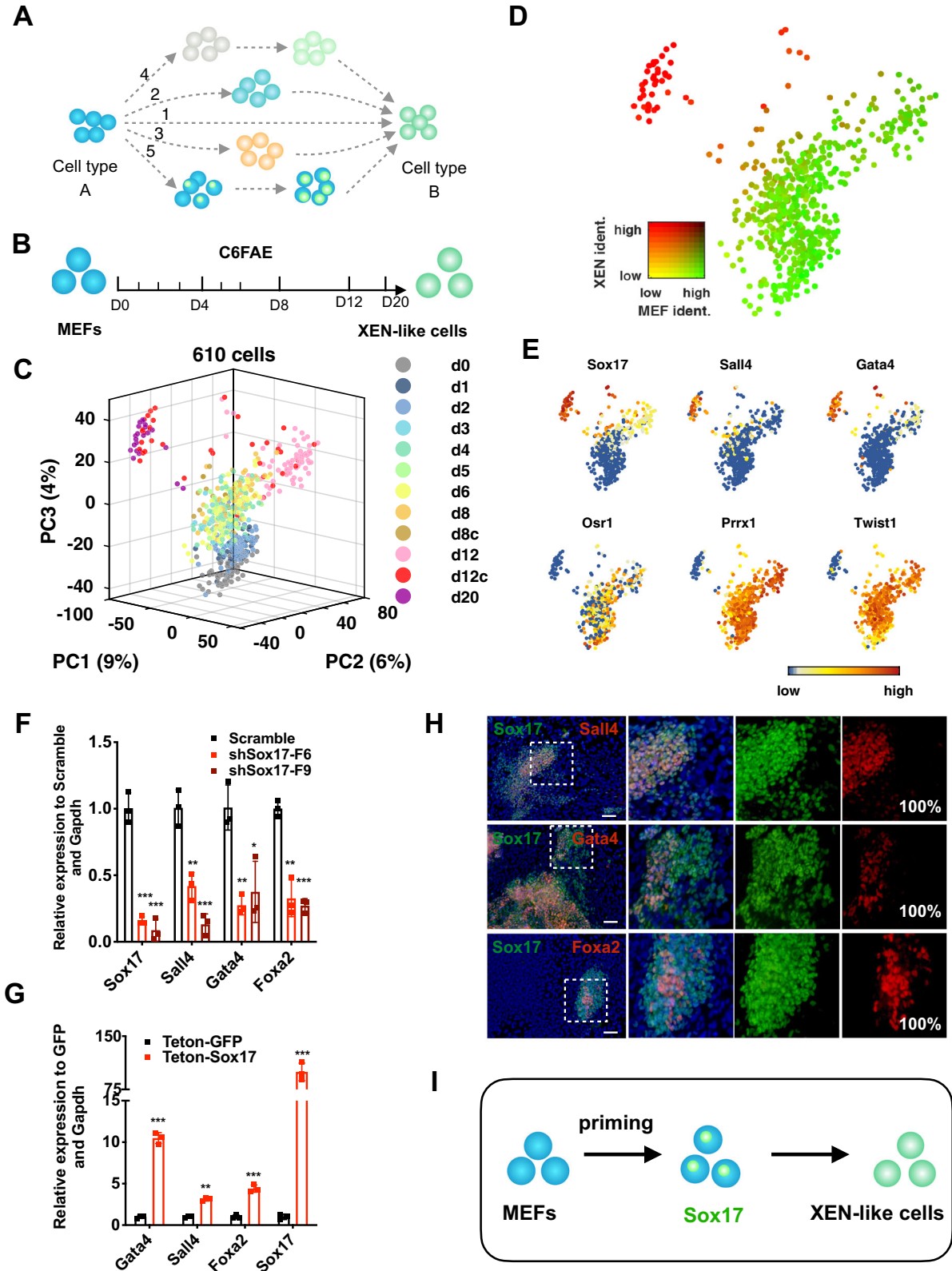

the number of Gata4-positive colonies greatly increased while the number of Sall4-positive colonies declined (Fig. 2c). Finally, at day 12, Sall4-positive cells turned out to be a subpopulation of Gata4-positive cells (Fig. 2a). This was probably due to the self-repression function of Sall4 expression as reported[32] or resulted from another wave of Gata4 activation without Sall4. Staining for Foxa2 revealed a subpopulation of Gata4 expressing cells, leading

us to believe that Gata4 might be upstream of Foxa2 in cell fate specification (Fig. 2d).

In a knockdown experiment of *Sall4* the expression of *Gata4* and *Foxa2* was severely disrupted, suggesting that *Sall4* is the upstream regulator of *Gata4* and *Foxa2*, which is consistent with the immunostaining data (Fig. 2e). Knockdown of *Gata4* decreased the expression of *Foxa2* but had no influence on the

**Fig. 1 Chemically induced *Sox17* expression initiates cell fate reprogramming towards XEN-like cells. a** Potential models for cell fate reprogramming. The "directly switch model" proposes cell fate switch directly without any intermediate cell type (1); The "de-differentiate and re-differentiate model" indicates cell fate reprogramming mediated by a multipotent stem cell with the differentiation potential into both the initial and target cell types (2); The "discrete state transit model" assumes cell fate reprogramming process with gradual fading of the initial cell features and gradual formation of the target cell identities (3); The "reset and reconfigure model" refers to cell fate reprogramming with the erasing of initial cell identity before the establishment of target cell identity (4); The "prime, specify and transit model" indicates cell fate reprogramming with priming and specification state without substantial alteration of initial cell identities before cell fate transition into the target cell types (5). **b** Schematic diagram of chemical-mediated XEN-like cell reprogramming. **c** PCA projection of all individual cells during the reprogramming process. d8c, single cells picked from colonies of day 8; d12c, single cells picked from colonies of day 12; **d** MEF and XEN identity in the PCA projection. For each cell on the XEN reprogramming path, the similarity to bulk RNA-seq from either MEFs or XEN-like cells as calculated using quadratic programming. **e** Expression of XEN master genes (*Sox17, Sall4, Gata4*) and MEF master genes (*Osr1, Prrx1, Twist2*) in the PCA projection. **f** Relative mRNA levels of XEN genes induced by C6FAE on day 12 with the knockdown of *Sox17* ($n = 3$). **g** Relative expression of XEN genes induced by C6FAE on day 12 with the overexpression of *Sox17* ($n = 3$). **h** Co-staining of Sox17 and other XEN master genes induced by C6FAE on day 12. Scale bar, 100 μm. The percentage of Sall4, Gata4 and Foxa2-positive cells emerged in Sox17-positive cells were labeled in the lower right corner of each picture. **i** Schematic of the stepwise XEN induction mediated by *Sox17* activation. Significance was assessed compared with the controls using a one-tailed Student's *t*-test. ***$p < 0.001$; **$p < 0.01$; *$p < 0.05$.

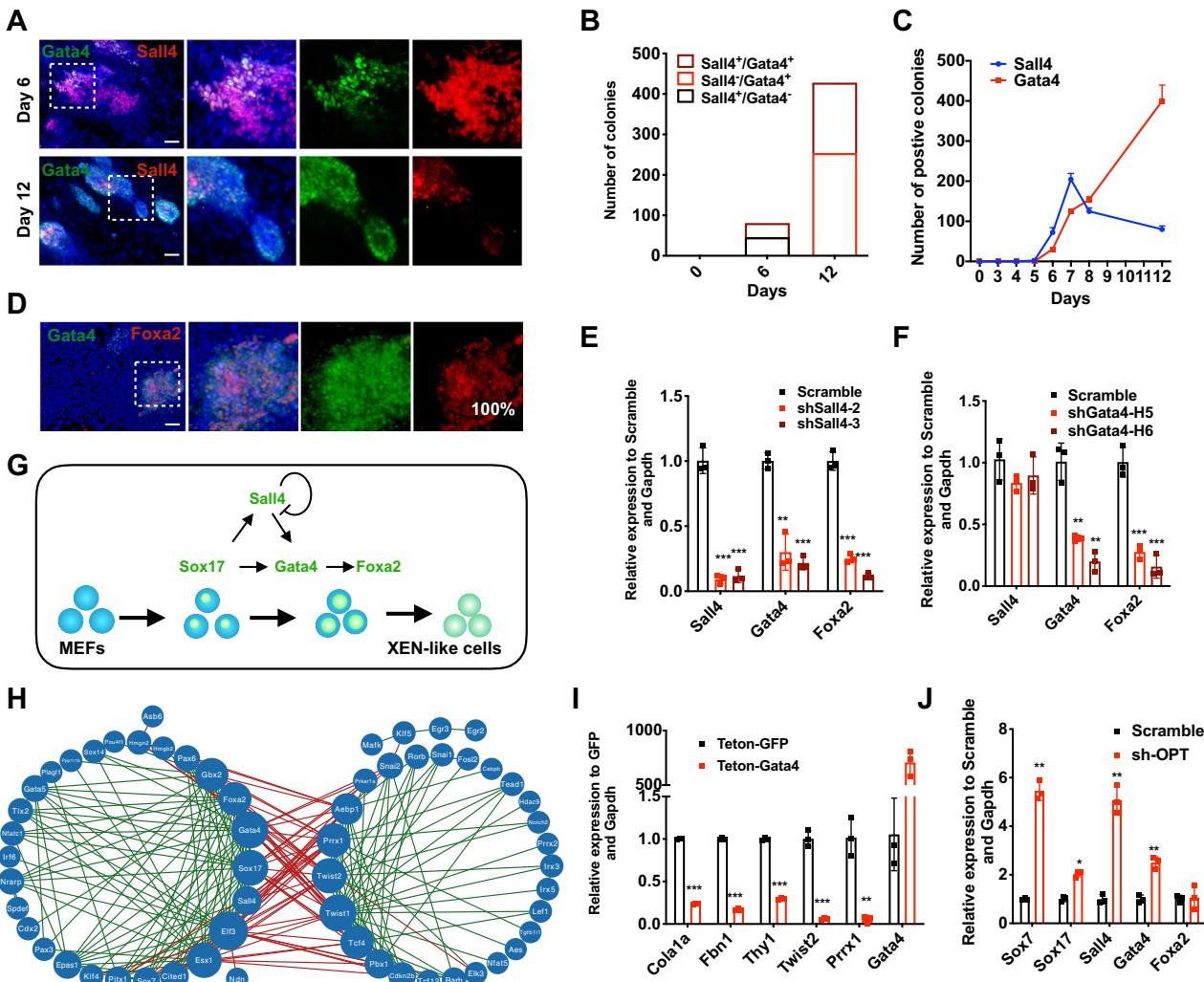

**Fig. 2 Cell fate specification and transition into XEN-like cells with the accumulated expression of master TFs after *Sox17* activation. a** Co-staining of Sall4 and Gata4 induced by C6FAE on day 6 and day 12. Scale bar, 100 μm. **b** Quantitation of Sall4$^+$/Gata4$^+$, Sall4$^+$/Gata4$^-$, Sall4$^-$/Gata4$^+$ colonies per well of 12-well plate induced by C6FAE on day 0, 6, and 12. **c** Numbers of Sall4 or Gata4-positive colonies per well of 12-well plate at different time points. **d** Co-staining of Gata4 and Foxa2 induced by C6FAE on day 12. Scale bar, 100 μm. The percentage of Foxa2-positive cells emerged in Gata4-positive cells were labeled in the lower right corner of the picture. **e** Relative mRNA levels of XEN genes induced by C6FAE on day 12 with the knockdown of *Sall4* ($n = 3$). **f** Relative mRNA levels of XEN genes induced by C6FAE on day 12 with the knockdown of *Gata4* ($n = 3$). **g** Schematic diagram of the hierarchical regulation circuitry among XEN master genes. **h** Transcription factor correlation network of XEN-like cells and MEFs. Green lines represent positive correlation and red lines represent the negative correlation. **i** Relative mRNA levels of MEF master genes with the overexpression of *Gata4* ($n = 3$). **j** Relative mRNA level of XEN genes with the knockdown of MEF genes. Sh-*OPT* stands for triple knockdown of MEF master genes, *Osr1, Prrx1*, and *Twist2* ($n = 3$). Significance was assessed compared with the controls using a one-tailed Student's *t*-test. ***$p < 0.001$; **$p < 0.01$; *$p < 0.05$.

transcription of *Sall4* (Fig. 2f), which further supported that Foxa2, but not Sall4, is a downstream factor of Gata4. In summary, we found the activation of *Gata4* and *Sall4* was regulated differently, and the mutual regulation between them was dynamic. Figure 2g shows the hierarchical regulatory network of XEN master TFs.

The regulatory network for the cell fate specification and the transition was established after the core network of XEN master TFs was constructed (Fig. 2h). The cell fates of MEF and XEN-like were seen mutual antagonizing from the regulatory network. In the transition process, the up-regulation of XEN master TFs promoted the down-regulation of fibroblast master TFs, and vice versa (Fig. 2i, j and Supplementary Fig. 2). Such a positive feedback loop could account for the fast transition from fibroblasts to XEN-like cell fates in the final stage of reprogramming, which was exhibited in the single-cell analysis (Fig. 1f and Supplementary Fig. 1h).

Taken together, the core XEN transcriptional network, including *Sox17*, *Sall4*, *Gata4*, and *Foxa2*, was established consecutively and hierarchically, and thus completing the cell fate specification and transition process in the end. These findings supported the "prime, specify and transit" model that previously speculated (model 5 in Fig. 1a).

**Chemicals are essential for *Sox17* expression while play different roles in the specification and transition process.** Since CHIR99021, 616452 and Forskolin are pivotal to chemical reprogramming to XEN-like cells (Supplementary Fig. 3a), we further explored whether they were necessary to the entire process of reprogramming. We found that the chemical compounds worked with a stepwise approach.

Subtracting any one of CHIR99021, 616452 and Forskolin, respectively, from C6FAE from day 0 hampered the expression of XEN master TFs, especially the major TF, *Sox17* (Supplementary Fig. 3b). We then withdrew CHIR99021, 616452 and Forskolin after 4-day induction when *Sox17* was already activated albeit at a lower level (Fig. 3a). We found that the presence of CHIR99021 and Forskolin was essential for *Gata4* activation, while the addition of 616452 was critical for the up-regulation of *Sall4* (Fig. 3a). The expression of *Foxa2* was also greatly impaired when CHIR99021 or Forskolin was removed (Fig. 3a), which is consistent with our previous finding that *Foxa2* activation might be downstream of *Gata4* activation.

We further studied the requirement of CHIR99021, 616452, and Forskolin for the protein expression of Sall4 and Gata4, by subtracting CHIR99021, 616452, and Forskolin after day 6, when Sox17-positive cell number was greatly increased. Similar to the transcriptional level, 616452 was essential for the expression of Sall4 protein, and chemical cocktails containing 616452 after 6-day treatment of C6FAE were sufficient to induce the expression of Sall4 protein (Fig. 3b, c). Moreover, we detected the expression of Gata4 in Sall4-positive colonies when CHIR99021 or Forskolin was subtracted from the cocktail after day 6 in the presence of 616452 (Fig. 3b, c). It was consistent with our previous findings that Sall4 activated Gata4 expression. Interestingly, when 616452 was removed from the cocktail after day 6, in the presence of CHIR99021 and Forskolin, Gata4 expression was still detected at a high level, and Sall4 expression was substantially impaired. This indicates CHIR99021 and Forskolin were sufficient to induce Gata4 expression after the activation of Sox17, which is independent of Sall4 expression (Fig. 3d, e). This was also consistent with another wave of Gata4 expression that was found after 6 days of C6FAE treatment (Fig. 2a–c).

In summary, it was the cooperation of CHIR99021, 616452 and Forskolin that activated *Sox17*; thereafter, CHIR99021/Forskolin

and 616452 activated the expression of *Gata4* and *Sall4*, respectively, in the specification stage, which further established the entire core regulatory network of XEN (Fig. 3f). After day 8, CHIR99021, 616452, and Forskolin were not essential for XEN gene expression (Supplementary Fig. 3c), suggesting that the transition phase was a self-organizing process by XEN master genes. These were also in line with the findings that the transduction of *Sall4* and *Gata4* was able to replace the function of CHIR99021, 616452 and Forskolin in inducing XEN-like colonies[4]. Overall, CHIR99021, 616452, and Forskolin played different roles in the reprogramming processes before and after Sox17 expression although they were required for both of the two phases.

**Endogenously activated BMP signaling is critical for *Sox17* activation and XEN induction.** We further explored the upstream factors of *Sox17* after chemical treatment. Using bulk RNA-sequencing in the very early stage of XEN reprogramming, we found that *Bmp2* was one of the factors that were activated before *Sox17* expression (Fig. 4a and Supplementary Fig. 4a).

To investigate the roles of *Bmp2* in the activation of *Sox17* and the subsequent reprogramming into XEN-like cells, we inhibited Bmp signaling with small molecule inhibitors Dorsomorphin and DMH1. We found that both the transcription of *Sox17* and the number of Sox17-positive colonies remarkably decreased (Fig. 4b–d). Meanwhile, the overexpression of *Bmp2* promoted the activation of Sox17 drastically (Supplementary Fig. 4b–e). Adding recombinant BMP2 or BMP4 in the reprogramming medium also improved the messenger RNA (mRNA) level of Sox17 and the number of Sox17-positive colonies (Fig. 4e–g). Consistently, Dorsomorphin and DMH1 compromised the upregulation of Sox17 expression by BMP2 or BMP4 (Supplementary Fig. 4f–i).

Importantly, the mRNA level of other XEN master genes (*Sall4*, *Gata4*, *Foxa2*) and the efficiency of XEN-like cell induction were hampered by Dorsomorphin and DMH1 (Fig. 4h, i), and were promoted by BMP2 and BMP4 (Fig. 4j, k). Also, we found that BMP4 notably promoted the up-regulation of *Sox17* in the iCD1 serum-free medium used in CiPSC induction[9] (Supplementary Fig. 4j). However, the effects of BMP4 on the activation of *Sox17* relied on the presence of C6F. BMP4 could not replace the role of C6F on the activation of *Sox17* (Supplementary Fig. 4k).

These results indicate that the early activation of endogenous Bmp signaling by chemical cocktails promoted the expression of *Sox17* and thus facilitated the stepwise induction of XEN-like cells (Fig. 4l and Supplementary Fig. 4l).

**The chemical boosters, CH55 and VPA, benefit *Sox17* activation and XEN specification differently.** The two phases before and after Sox17 expression revealed in our study, raised the possibility that the chemical boosters played different roles in the stepwise process from fibroblast to XEN-like cells. Thus, we compared the gene expression profiles induced with and without the previously reported chemical boosters, CH55 and valproic acid (VPA), in the presence of C6FAE[3,5]. Interestingly, CH55 promoted the expression of *Sox17* notably in the first 4 days, even on the basis of exogenously provided Bmp4 (Supplementary Fig. 5a), while had nearly no function in further activation of other XEN genes from day 4 to 12 (Fig. 5a, c). VPA was found to promote the up-regulation of most XEN genes and XEN identity from day 4 to day 12 (Fig. 5b, c). However, VPA has no beneficial effect on *Sox17* expression in the first 4 days, suggesting that VPA improved XEN reprogramming efficiency by supporting the up-regulation of the XEN network after the activation of endogenous Sox17.

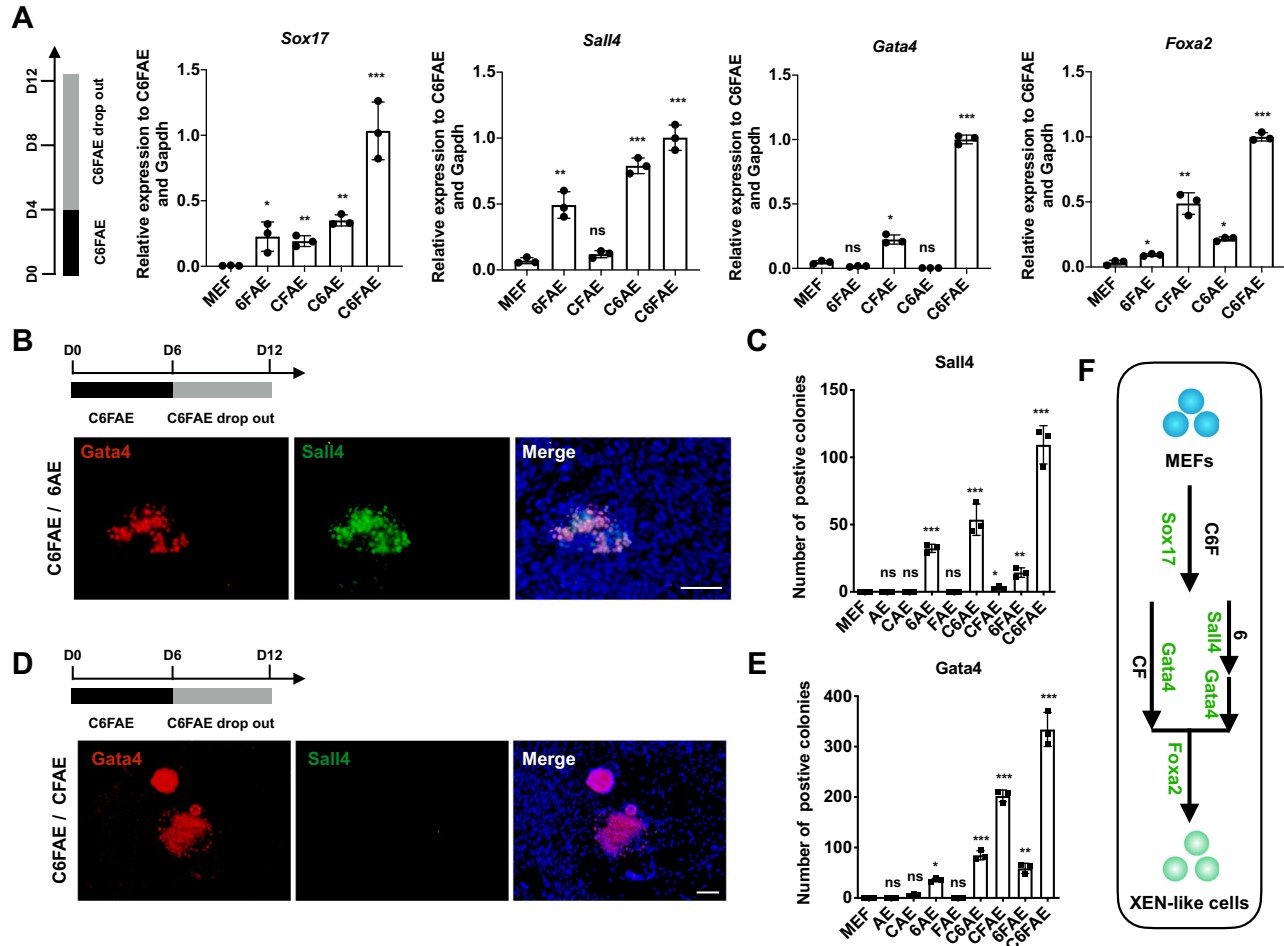

**Fig. 3 CHIR99021, 616452, and Forskolin are essential for *Sox17* expression while play different roles in the specification and transition process.**
**a** Relative mRNA levels of key XEN master genes on day 12 with removing C, 6, and F, respectively, from day 4 ($n = 3$). **b** Immunostaining of Sall4 and Gata4 on day 12 after removal of C, F from day 6. Scale bar, 100 μm. **c** Sall4-positive colony numbers on day 12 after removal of C, 6, and F, respectively, from day 6 ($n = 3$). **d** Immunostaining for Sall4 and Gata4 on day 12 after removal of 6 from day 6. Scale bar, 100 μm. **e** Gata4-positive colony numbers on day 12 after removal of C, 6, and F, respectively, from day 6 ($n = 3$). **f** Schematic of the roles of C, 6, and F in the regulation of XEN master genes. Significance was assessed compared with the controls using a one-tailed Student's $t$-test. ***$p < 0.001$; **$p < 0.01$; *$p < 0.05$.

We also found that the cocktail mainly induced "smoothed" colonies co-expressing *Sox17*, *Gata4*, and *Sall4* in the presence of VPA (VC6FAE). Without VPA, it induced many "fuzzy" colonies with robust *Sox17* expression and very low expression of *Sall4* and *Gata4* (Supplementary Fig. 5b–d). The fuzzy colonies had higher mRNA levels of the fibroblast master genes, *Osr1*, *Prrx1* and *Twist2* (Supplementary Fig. 5e) and could rarely be induced into XEN-like cells (Supplementary Fig. 5f, g). Importantly, smoothed colonies, but not fuzzy colonies, could be induced into pOct4-GFP-positive CiPSCs (Supplementary Fig. 5h, i). VPA promoted the induction of pOct4-GFP-positive CiPSCs (Supplementary Fig. 5j). These results support that VPA improves the C6FAE-mediated XEN reprogramming by promoting the XEN specification process, which was previously reported to bridge chemical reprogramming from fibroblasts to pluripotency[4,5].

We further determined whether using chemical boosters, CH55 and VPA, in a stepwise manner could promote the XEN reprogramming efficiency. We found that treating the cells with CH55 only in the first 4 days was more efficient than treating for the entire process in promoting the expression of *Sox17* and *Sall4* (Fig. 5d). Also, VPA induced a higher level of *Gata4* and *Foxa2* mRNA when using in the last 8 days rather than in the entire process (Fig. 5e). Collectively, the sequential use of CH55 and

VPA in different steps reached the highest efficiency of XEN-like colony generation (Fig. 5f, g). Taken together, these findings not only suggested the "prime, specify and transit" model in chemical reprogramming but also revealed the roles of the chemicals on the stepwise processes (Fig. 1a, h).

## Discussion

A major question in chemical reprogramming is "how does a set of chemicals, which bear no obvious relation to any genes or molecules that are directly associated with a specific cell type, enable the determination of a specific cell fate". In this study, we made a significant conceptual leap towards an answer to this question. We demonstrated that the chemical reprogramming was a stepwise process by studying the molecular roadmap from fibroblasts to XEN-like cells. First, the chemicals orchestrated a priming state with the activated expression of Sox17, a master gene of XEN, without substantial cell fate alteration or determination. Afterward, the chemicals further guided hierarchical accumulation of endogenous master transcription factors for cell fate specification. Finally, cell fate was transited with the combination of those activated master transcription factors. In brief, chemicals used in reprogramming guided the hierarchical

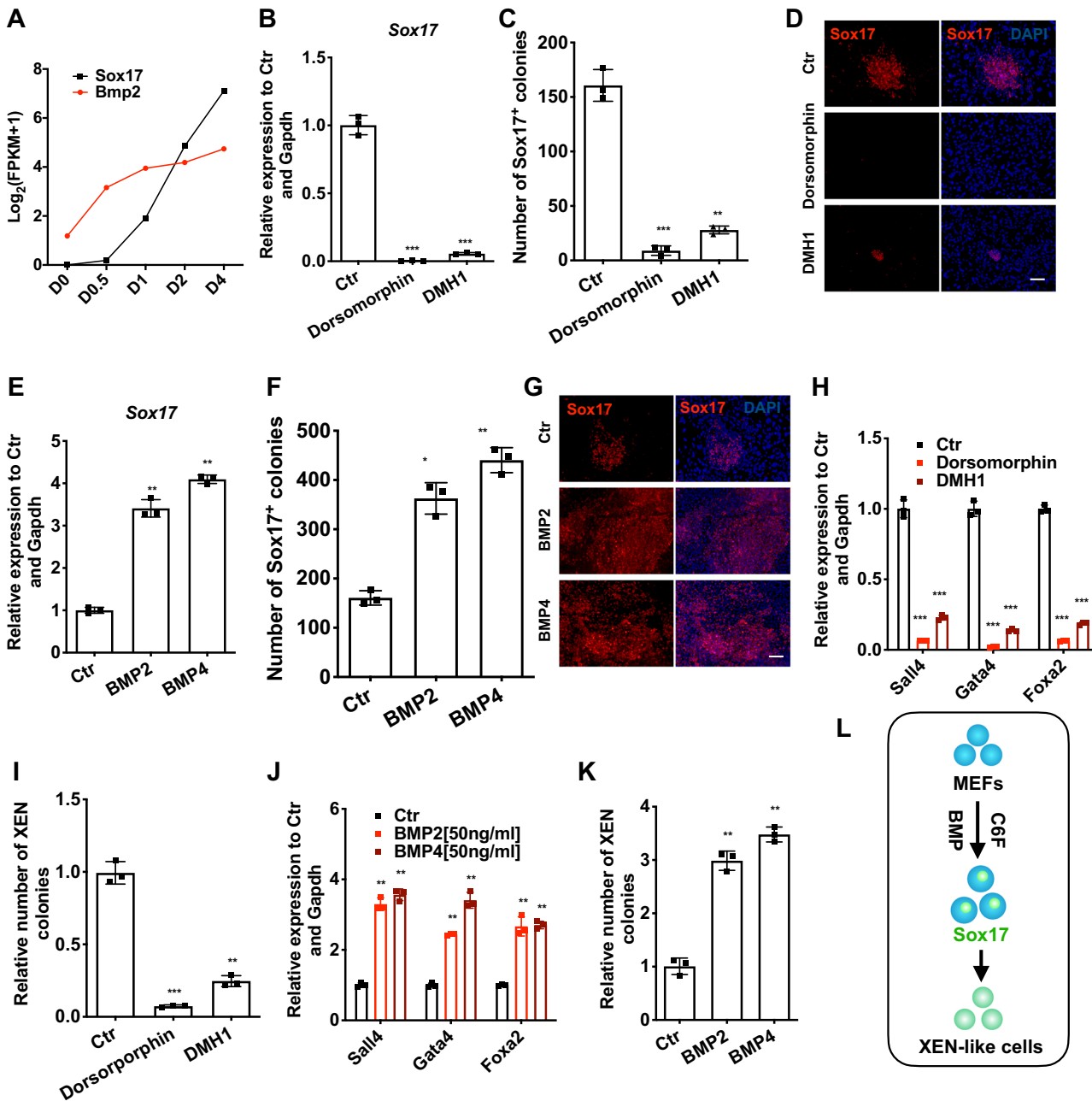

**Fig. 4 Endogenously activated BMP signaling pathway is critical for the activation of *Sox17* and XEN induction. a** Expression of *Bmp2* and *Sox17* at different time points (analyzed by bulk RNA-sequencing). **b** The effects of Dorsomorphin and DMH1 on C6FAE-mediated *Sox17* activation (analyzed by RT-qPCR on day 4) ($n = 3$). **c** Number of Sox17-positive cells induced with Dorsomorphin and DMH1 on day 6 ($n = 3$). **d** Sox17-positive cells induced with Dorsomorphin and DMH1 on day 6. Scale bar, 100 μm. **e** Relative expression of *Sox17* after 4 days treatment with BMP2 and BMP4 at the dosage of 50 ng/ml ($n = 3$). **f** Number of Sox17-positive cells induced with BMP2 and BMP4 on day 6 ($n = 3$). **g** Sox17-positive cells induced with BMP2 and BMP4 on day 6. Scale bar, 100 μm. **h** Relative expression of *Sall4*, *Gata4*, *Foxa2* induced with Dorsomorphin and DMH1 on day 6 ($n = 3$). **i** Relative number of XEN colonies induced with Dorsomorphin and DMH1 on day 12 ($n = 3$). **j** Relative expression of *Sall4*, *Gata4*, *Foxa2* induced with BMP2 and BMP4 on day 6 ($n = 3$). **k** Relative number of XEN colonies induced with BMP2 and BMP4 on day 12 ($n = 3$). **l** Schematic of the roles of BMP signaling in the regulation of *Sox17* and XEN cell fate. Significance was assessed compared with the controls using a one-tailed Student's *t*-test. ***$p < 0.001$; **$p < 0.01$; *$p < 0.05$.

activation of master genes in the cell-fate-associated regulatory network in a stepwise manner.

Therefore, we indicate that the chemicals previously used in the entire process from fibroblasts to XEN-like cells had different functions in different phases and played different roles in activating different genes. The core chemicals CHIR99021, 616452, and Forskolin (C6F) were all essential to stimulate *Sox17* expression in the priming phase. Afterward, they supported the activation of other master genes, such as *Sall4* and *Gata4* for the

XEN cell fate specification in the *Sox17* expressing cells differently. CHIR99021 and Forskolin facilitated *Gata4* expression, while 616452 enabled the expression of *Sall4*. CH55 and BMP signaling functioned through elevating *Sox17* activation, while VPA functioned through activating the other XEN master genes in the *Sox17* expressing cells.

Importantly, the "prime–specify–transit" model may be extended to other chemical reprogramming systems according to the gene expression profiling data during the reprogramming

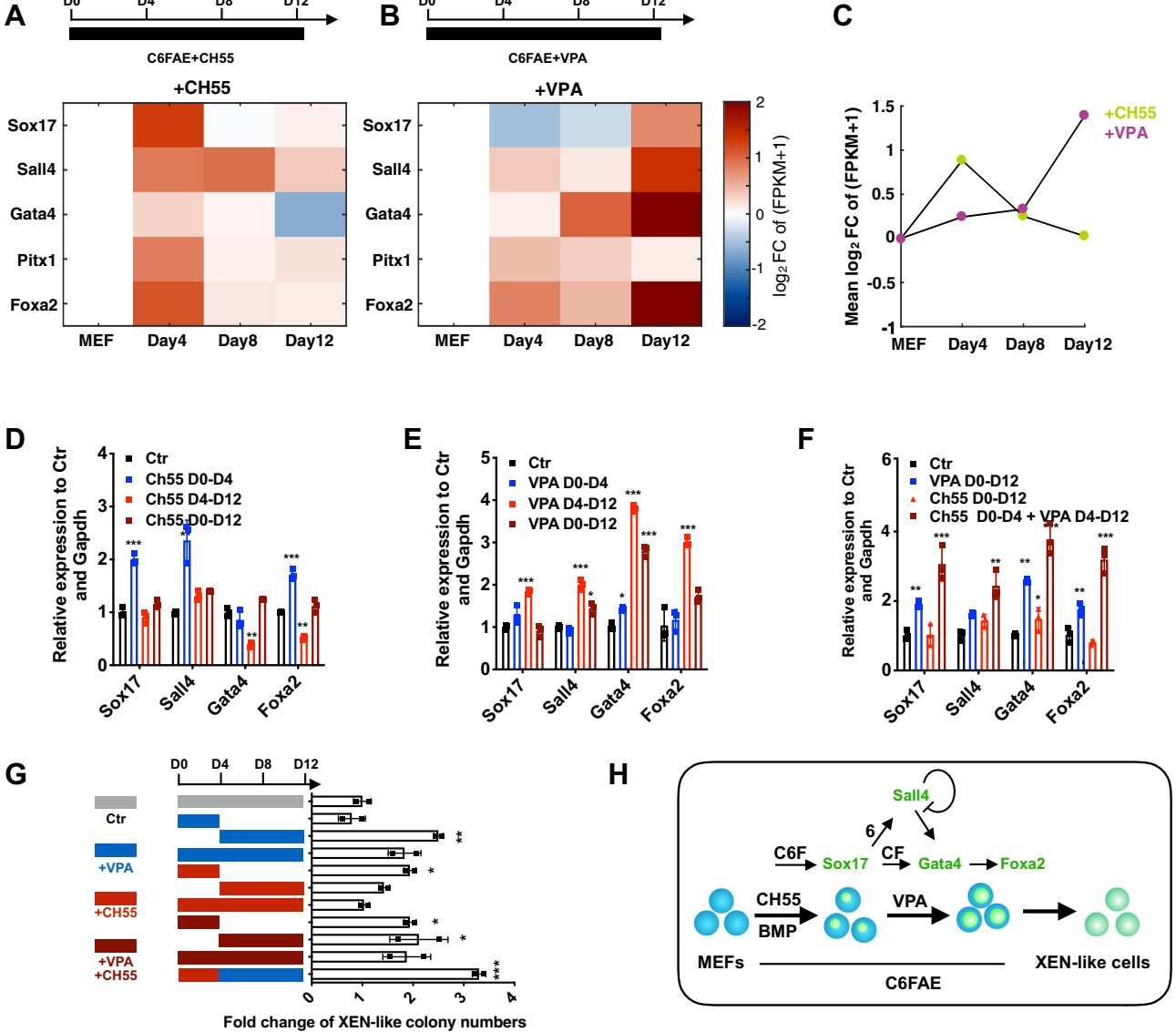

**Fig. 5 Chemical reprogramming boosters benefit different steps of reprogramming. a**, **b** Effects of CH55 (**a**) and VPA (**b**) on the expression of XEN genes at different time points. C6FAE with additional CH55 or VPA conditions were compared with only C6FAE condition. **c** Effects of CH55 and VPA on XEN identity at different time points. C6FAE with additional CH55 or VPA conditions were compared with only C6FAE condition. **d**, **e** Relative mRNA expression of XEN master genes in the presence of CH55 (**d**) or VPA (**e**) with different durations ($n = 3$). **f** Relative fold change of the XEN genes expression in cells treated with CH55 and VPA by different duration ($n = 3$). (analyzed on day 12 by RT-qPCR). **g** Relative numbers of XEN colonies in the presence of CH55 and VPA with different durations ($n = 3$). **h** Summary of the functions of chemical boosters, CH55 and VPA, in XEN induction. Significance was assessed compared with the controls using a one-tailed Student's $t$-test. ***$p < 0.001$; **$p < 0.01$; *$p < 0.05$.

processes. For instance, in the chemical reprogramming process from fibroblast to neural stem cells, *Sox2* was activated very early by small molecules in the first 4 days and might initiate a priming state to neural stem cell[33]. After that, neural stem cell core regulators network was built up, which was reminiscent of a specification process. Besides, the molecular dynamics during the chemical reprogramming from fibroblast to photoreceptor-like cells (CiPCs) was probably initiated by the activation of photoreceptor-specifying transcription factors, such as *Rorb*, *Ascl1* and *Pias3*, reminiscent of a priming phase[20]. The stepwise manner in the activation of the master transcription factors suggested identifying the key molecular events in a chemical reprogramming process could help to optimize the protocol in a stepwise manner to achieve higher efficiency.

Unexpectedly, although cells in the priming phase had already expressed some of the XEN master TFs, they were still fibroblast-

like with a high level of the fibroblasts program, as well as the high expression of fibroblast master genes. This was rather different from an intermediate multipotent cell type that was presumed in previous reports[34,35] and from the other possible intermediates that were speculated before this study (Fig. 1a). The "Disc model" for cell fate reprogramming matches these findings since cell fate priming helps the cell to escape the attractor of an initial cell type without cell fate determination, while the hierarchically accumulated expression of endogenous transcription factors provides the "guide rail" to determine a cell fate progressively, without entering into a multipotent attractor[36].

Our findings also highlight the importance of activating some or even one master transcription factor of the target cell type in developing a chemical reprogramming system. In the reprogramming process to XEN-like cells, the activation of Sox17 was a molecular event that was not easy to be triggered. It required

most chemicals in the cocktail like C, 6, F and CH55, and even took advantage of the endogenously activated expression of BMP2 or other BMP signaling stimuli from serum or KSR. Thereafter, Sox17 expression made the subsequent molecular events possible and easier. Thus, the activation of one or more transcription factors in a cell type may represent the major molecular basis for cell plasticity. The cells initially express one or more transcription factors of another cell type may have superiority in cell fate transition in chemical reprogramming.

Moreover, since the chemicals C, 6, F, and their combinations have been widely used in different chemical reprogramming systems and generate many different cell fates[8,12,13,15–17,19,20], it is still unclear whether Sox17 activation is a specific outcome of the chemical treatment and whether these chemicals can prime the cells and facilitate cell fate conversion into other lineages simultaneously. These are some of the questions we intend to address in our future study.

## Methods

**MEF isolation**. MEFs were isolated from E13.5 embryos of ICR mouse. After the removal of head, limbs, and viscera, embryos were minced with scissors and dissociated in trypsin-EDTA at 37 °C for 10 min. After centrifugation, MEF cells were collected and cultured in MEF medium, which included: high-glucose Dulbecco's modified Eagle's medium (DMEM) supplemented, 10% fetal bovine serum (FBS), 1% GlutaMAX, 1% nonessential amino-acids (NEAAs), and 1% penicillin-streptomycin. Oct4-EGFP mice were obtained from The Jackson Laboratory (004654). This study was performed under in accordance with protocols by Peking University laboratory animal research center.

**Generation of XEN-like cells from fibroblasts**. Twenty-thousand MEF cells were seeded into a well of 12-well plate. Twenty-four hours later later, the medium was changed to XEN reprogramming medium, which included: KnockOut DMEM supplemented, 10% KnockOut Serum Replacement (KSR), 10% FBS, 1% GlutaMAX, 1% NEAAs, 0.055 mM 2-mercaptoethanol, 1% penicillin–streptomycin (Invitrogen), 50 ng/ml basic fibroblast growth factor (bFGF), and the small-molecule cocktail VC6FAE (0.5 mM valproic acid, 20 μM CHIR99021, 10 μM 616452, 50 μM Forskolin, 0.05 μM AM580, and 5 μM EPZ004777). XEN reprogramming medium was changed every 4 days for 12 to 20 days.

**Immunofluorescence**. Primary antibodies were those specific to rabbit anti-SALL4 (Abcam, 1:500), goat anti-SOX17 (R&D, 1:500), goat anti-GATA4 (Santa Cruz, 1:300), goat anti-GATA6 (R&D, 1:200), rabbit anti-Nanog (Sigma Aldrich, 1:200). The secondary antibodies used were FITC-conjugated secondary antibodies and TRITC-conjugated secondary antibodies (Jackson ImmunoResearch, 1:200).

Cells were fixed in 4% paraformaldehyde for 15 min at room temperature. Then, removing 4% paraformaldehyde and washing cells with PBS for two times. cells were permeabilized and blocked in PBS containing 0.2% Triton X-100 and 3% donkey serum for 1 h at room temperature. Then the cells were incubated with primary antibodies at 4 °C overnight. After washing three times with PBS, secondary antibodies (Jackson ImmunoResearch) were incubated at 37 °C for 1 h. The nuclei were stained with DAPI (Roche Life Science) for 5 min.

**Quantitative reverse transcription PCR (RT-qPCR)**. RT-qPCR was performed according to protocols. Briefly, total RNA samples were extracted by using the EasyPure RNA Kit (TransGen Biotech) and were reverse transcribed into complementary DNA (cDNA) using TransScript One-step gDNA Removal and cDNA Synthesis SuperMix (TransGen Biotech); Real-time PCR was performed on a Quantagene q225 System (KUBO technology) using 2 × T5 Fast qPCR Mix (TSINGKE Biological Technology).

**Single-cell RNA-seq**. Individual cell at different time points was manually picked after digestion, lysed and subjected to cDNA synthesis[37,38]. Single-cell cDNA was then amplified and fragmented as published steps[37,38]. The sequencing library was constructed (New England Biolabs) and sequenced with paired-end 150-bp reads on an Illumina HiSeq X-Ten platform (Novogene). Raw reads were firstly separated by cell barcodes, then trimmed, and aligned to the mm9 mouse transcriptome and de-duplicated by UMIs information as described previously[39].

**Pseudotime analysis**. Monocle (v2.6.4) were adopted to perform the pseudotime analysis. Differentially expressed genes (DEGs) identified from each cell type were used as ordering genes. The whole workflow followed the recommended pipeline with default parameters.

**Integration analysis of gene expression between data in this study and in Zhao et al.**[6]. Cells from day 0 to day 20 in this study, and cells belong to MEFs, Stage I day 5, Stage I day 12 and XEN-like cells in Zhao et al.[6] were used to perform the integration analysis. To integrate different data sets, CCA algorithm from Seurat was used.

**Statistics and reproducibility**. All experiments contain at least three independent biological replicates. No randomization or blinding was used. The statistical analysis in this paper uses Student's t-test. p-value of <0.05 is considered a significant difference.

**Reporting summary**. Further information on research design is available in the Nature Research Reporting Summary linked to this article.

## Data availability

The accession number for the RNA-seq and single-cell RNA-seq data reported in this paper is NCBI GEO: GSE144097. Source data underlying plots shown in figures are provided in Supplementary Data 1. Full blots are shown in Supplementary Information. All other data, if any, are available upon reasonable request.

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

## Acknowledgements

We thank Chunyan Yang, Yang Liu, and Siyuan Zhang for technical assistance, and Jiayu Chen for providing pOct4-GFP mice. We thank Iain C. Bruce for editing the manuscript. This work was supported by the National Key Research and Development Program of China (2018YFA0800504), the National Natural Science Foundation of China (31771645, 31922020, 31821091 and 31771590), the Science and Technology Department of Sichuan Province (2018JZ0025).

## Author contributions

Z.Y., J.L., Q.W., C.Y., L.M., J.Y., and K.B. performed experiments. X.X., C.G., and A.N. conducted the bioinformatics analyses. Y.Z., C.T., and F.G. supervised this project. Y.Z., Z.Y., and X.X. wrote the paper.

## Competing interests

The authors declare no competing interests.
