## [Peer Review File · Communications Biology]

Reviewers' comments:

Reviewer #1 (Remarks to the Author):

It is important and essential to figure out exact mechanisms of chemical modulation of cell fate transition during chemical reprogramming. In this manuscript, Yang and colleagues showed chemical induced master transcription factors activation dynamics during fibroblasts translated into XEN cells process. The authors found that three chemicals C6F is necessary for Sox17 activation at day4, which further trigger core network regulators of XEN cells like Sall4, Gata4 and Foxa2 expression. Additionally, they stated that Sox17 activation depended on BMP signaling and optimized chemicals treatment like supplied boosters CH55 and VPA could produce two different XEN colonies. Though interesting, it is necessary to provide more solid evidence to support statement mentioned here, which is showed as follows.

Major concerns:

As the author shown, CHIR99021, 616452 and Forskolin are necessary and sufficient to activate XEN master TFs, including Sox17, Sall4 and Gata4 in a stepwise manner. So can these factors replace the three chemicals to mediate the generation of XEN cells in a similar manner?

The author shown endogenous BMP signaling is critical for activating Sox17 and XEN induction. However, the chemical reprogramming protocol they adopted was serum-contained and BMPs are present in serum. It is better to evaluate the function of BMPs in a serum-free system. Besides, can BMP2/4 replace C6F to activate Sox17? And what is the mechanism of BMP signaling regulates Sox17? Directly or indirectly? In addition, the evidence of activated BMP signaling in the protein level such as WB or IF of p-smad1/5/8 is needed.

Here the author only showed that BMP signaling or CH55 regulated Sox17 activation respectively, whether any relationship or interaction model between BMP signaling and CH55 work for Sox17 activation? Additionally, what VPA and C6F?

What is difference between fuzzy and smoothed XEN colonies except some marker gene expression, like epigenetics or any influence on pluripotency induction in the later stage of CiPSC induction.

What function do chemicals mentioned here play in XEN cells self-renew ability?

Minor points :

Somewhere in the manuscript gene name should be italic.

Figure 1 legend: Figure 1K should be Figure 1I.

Reviewer #2 (Remarks to the Author):

How chemical cocktails orchestrate the cell fate alteration and determination during chemical reprogramming at the molecular levels is an interesting and important question, as these chemicals, in general, do not possess cell-type specificities like transcription factors. In this manuscript, Yang and colleagues sought to address this question by carefully tracing the reprogramming trajectory using scRNA-seq. In combination with genetic experiments, they dissected the relationship of these chemicals and the activation of master regulators such as Sox17, Sall4, Gata4, Foxa2. Furthermore, the authors found that BMP pathways activation is upstream of SOX17 in XEN-like cell induction. Finally, they elucidated the roles of two additional chemical boosters during reprogramming and their functional timing, which led to an improved reprogramming protocol. In general, this study, which is elegantly designed, carefully dissected the temporal order of molecular events underlying the chemically induced cell reprogramming. I found it illuminating, and the authors are appreciated for their efforts to generate these large-scale datasets and functional validation. The proposed "prime, specify and transit" model is also an interesting concept. I do have a few minor suggestions to help improve the paper.

1. The authors should introduce more background or supporting evidence for the XEN-like master factors circuitry (Sox17-Sall4-Gata4-Foxa2). This may not be immediately evident for readers outside of reprogramming field.

2. The authors need to explain C6FAE when first introduced, and why they chose to investigate C6F but left out AE function in this study.

3. The authors stated "we found that cells were quite close to each other during the earlier days and then gradually separated, at last dividing into two branches" - "the proceeding branch" (left branch) and "the trapped branch" (right branch). To me, most d20 cells are in the proceeding branch. Does it mean that d20 cells in the trapped branch largely died? Otherwise this may not be an accurate statement. I also suggest to perform a differential analysis between pseudo-time and the real-time of the RNA-seq data to better explain the trajectory.

4. The single-cell RNA-seq data require quality controls. The author may comment on the comparison with existing datasets along the chemical reprogramming process, for example, Zhao et al. (Cell Stem Cell, 2018).

5. Can the authors perform Sox17 ChIP-seq (if a good antibody is available) or luciferase assay to see whether Sox17 directly regulates the transcription of these downstream factors?

6. Can the authors discuss how likely this "prime-specify-transit" model may be applicable to other chemically induced reprogramming and reprogramming in general?

Minor comments:

7. Fig. 1A requires more detailed explanation. What is the difference between the "de-differentiate and re-differentiate model" and "prime, specify and transit model"?

8. Fig. 2A,D are a bit too small to read. Quantitative analysis should be provided for all IF results.

9. The authors stated that "endogenous expression of Sox17 is significantly activated by chemical cocktails in the first 4 days". This is also clear in Fig. S1E. However, this is not obvious in Fig. S1D. Since both figures are presumably from the same datasets, this needs to be clarified.

10. It would be nice if the authors can analyze the correlation of the Sox17 expression with Sall4, Gata4, and Foxa2 in the single cell level from their scRNA-seq.

11. This might be out of the scope of the paper, but in addition to BMP, can the authors discuss are there any potential new potential master regulators that are upstream of Sox17? This may shed lights on even earlier regulators.

12. The authors mentioned several manuscripts in submission in the paper. Unless they are accompanied by this submission (or BioRxiv deposition), I suggest to reduce such discussions of unpublished data which are difficult to judge without seeing the data.

Reviewer #1 (Remarks to the Author):

It is important and essential to figure out exact mechanisms of chemical modulation of cell fate transition during chemical reprogramming. In this manuscript, Yang and colleagues showed chemical induced master transcription factors activation dynamics during fibroblasts translated into XEN cells process. The authors found that three chemicals C6F is necessary for Sox17 activation at day4, which further trigger core network regulators of XEN cells like Sall4, Gata4 and Foxa2 expression. Additionally, they stated that Sox17 activation depended on BMP signaling and optimized chemicals treatment like supplied boosters CH55 and VPA

could produce two different XEN colonies. Though interesting, it is necessary to provide more solid evidence to support statement mentioned here, which is showed as follows.

Major concerns:

Q1. As the author shown, CHIR99021, 616452 and Forskolin are necessary and sufficient to activate XEN master TFs, including Sox17, Sall4 and Gata4 in a stepwise manner. So can these factors replace the three chemicals to mediate the generation of XEN cells in a similar manner?

Response 1: We agree with the reviewer that it is important to show whether those endogenously activated XEN factors can replace the three key chemicals. Actually, in the previous study, it has been reported that the overexpression of “Sall4 plus Gata4” or “Sall4 plus Gata6” in fibroblasts is sufficed in inducing XEN-like colony formation in the absence of the three key small molecules, CHIR99021, 616452, and Forskolin, as shown below (Response figure 1. or Figure 6D in Zhao et al., *Cell*, 2015. DOI: doi.org/10.1016/j.cell.2015.11.017). To better elaborate on this issue, we added a sentence in the revised version of manuscripts (page 12, line 8-10).

Response figure 1. Numbers of XEN-like colonies by overexpression of *SALL4* (*S4*), *GATA4* (*G4*), *GATA6* (*G6*), or their combinations in the presence of small-molecule cocktail VTAE (withdrawal of C6F from VC6TFAE) treatment. Figure 6D, Zhao et al., *Cell*, 2015. DOI: doi.org/10.1016/j.cell.2015.11.017.

Q2: The author shown endogenous BMP signaling is critical for activating Sox17 and XEN induction. However, the chemical reprogramming protocol they adopted was

serum-contained and BMPs are present in serum. It is better to evaluate the function of BMPs in a serum-free system. Besides, can BMP2/4 replace C6F to activate Sox17? And what is the mechanism of BMP signaling regulates Sox17? Directly or indirectly? In addition, the evidence of activated BMP signaling in the protein level such as WB or IF of p-smad1/5/8 is needed.

Response 2: We thank the reviewer for making us aware of this. We used serum-free iCD1 medium and tested the effect of BMP4 on the activation of *Sox17*. Consistently, we found that BMP4 promoted the up-regulation of *Sox17* in iCD1 serum-free medium (response figure 2-1, left). Although BMP4 promoted the activation of *Sox17*, it relied on the presence of C6F, so that BMP4 could not replace the role of C6F on the activation of *Sox17* (response figure 2-1, middle). We also detected that BMP4 elevated the phosphorylation level of Smad1/5/8, indicating that the BMP signaling was activated after the adding of BMP4 (response figure 2-1, right). We added the new data in Extended Data Fig. 4J-L.

Response figure 2-1. Left: Relative expression of *Sox17* after 6 days treatment with BMP4 on the basis of C6FAE in iCD1 serum-free medium. Middle: Relative expression of *Sox17* after 6 days treatment with BMP4 or C6F on the basis of AE. Right: Effect of BMP4 on the phosphorylation of Smad1/5/8 and the activation of *Sox17*. (analyzed on day 6 by western blot).

Although we have not determined whether BMP signaling directly or indirectly activate *Sox17* in chemical reprogramming, we hope the reviewer agree with us that it could be a negligible minor point in this report to mainly show the hierarchical and stepwise manner of endogenous transcription factor activation mediated by *Sox17*. To discuss more about this point, we checked some ChIP-seq data in the literature that the downstream factor of BMP signaling, Smad4, was able to bind directly to *Sox17* promoter and regulate its expression (response figure 2-2).

Response figure 2-2. Binding site of Smad4 on *Sox17* in mouse embryonic stem cells. ChIP-seq data was adapted from Martin-Malpartida et al., *Nat Commun*, 2070. doi: 10.1038/s41467-017-02054-6.

Q3: Here the author only showed that BMP signaling or CH55 regulated *Sox17* activation respectively, whether any relationship or interaction model between BMP signaling and CH55 work for *Sox17* activation? Additionally, what VPA and C6F?

Response 3: We appreciate the reviewer for pointing out this important aspect. To investigate the relationship between BMP4 and CH55 on *Sox17* activation, we combined BMP4 and CH55 together in the early phase of XEN reprogramming. As the histogram showed, CH55 further activated *Sox17* on the basis of BMP4 (response figure 3), which meant that CH55 and BMP signaling synergistically enhanced *Sox17* expression. The data was added in the revised manuscript (extended fig. 5A.)

In addition, according to our findings, C6F synergistically activated *Sox17* (Extended Data Fig. 3B). VPA promoted XEN reprogramming in the transition phase, but not in the priming phase (Fig.5), which was indicated by the activation of *Sox17*. So, CHIR99021, 616452 and Forskolin were all necessary for *Sox17* activation, but VPA played no role in the activation of *Sox17*.

Response figure 3. Relative expression of *Sox17* induced by CH55, BMP4 and CH55+BMP4 on the basis of C6FAE at day 4.

Q4. What is difference between fuzzy and smoothed XEN colonies except some marker gene expression, like epigenetics or any influence on pluripotency induction in the later stage of CiPSC induction.

Response 4: We had picked and passaged smoothed and fuzzy colonies for CiPSCs induction, and found that smoothed colonies, rather than fuzzy colonies, could be induced into Oct4-GFP positive CiPSCs (response figure 4, left and middle). And VPA promoted the induction of Oct4-GFP positive CiPSC colonies (response figure 4, right). So, we thought that smooth colonies had stronger CiPSCs induction ability than fuzzy colonies. This result was added in the revised version of our manuscript (Extended Data Fig. 5G-I).

Response figure 4. Left: Cell induction with CiPSC induction medium from fuzzy and smoothed colonies; Middle: Number of Oct4-GFP positive colonies induced from fuzzy and smoothed colonies; Right: Number of Oct4-GFP positive colonies induced by C6FAE and VC6FAE.

Q5: What function do chemicals mentioned here play in XEN cells self-renew ability?

Response 5: It could be an interesting question to study the function of the chemicals in regulating the self-renewal of XEN-like cells. We treated the chemically-induced XEN-like cell products with cocktails subtracting each small molecule for 8 days. We found that the dropout of 616452 only mildly affected the proliferation and clone formation of XEN, while CHIR99021 and Forskolin were essential to the self-renewal of XEN-like cells. We hope the reviewer agree with us that this point is out of the scope of this manuscript, so that we would not add this data in the revised manuscript.

Response figure 5. The growth curve of XEN colonies with cocktails subtracting each small molecule.

Minor points:

Somewhere in the manuscript gene name should be italic.

Figure 1 legend: Figure 1K should be Figure 1I.

Response: We are sorry for these mistakes. We had already adjusted these parts in our manuscript and figures.

Reviewer #2 (Remarks to the Author):

How chemical cocktails orchestrate the cell fate alteration and determination during chemical reprogramming at the molecular levels is an interesting and important question, as these chemicals, in general, do not possess cell-type specificities like transcription factors. In this

manuscript, Yang and colleagues sought to address this question by carefully tracing the reprogramming trajectory using scRNA-seq. In combination with genetic experiments, they dissected the relationship of these chemicals and the activation of master regulators such as Sox17, Sall4, Gata4, Foxa2. Furthermore, the authors found that BMP pathways activation is upstream of SOX17 in XEN-like cell induction. Finally, they elucidated the roles of two additional chemical boosters during reprogramming and their functional timing, which led to an improved reprogramming protocol. In general, this study, which is elegantly designed, carefully dissected the temporal order of molecular events underlying the chemically induced cell reprogramming. I found it illuminating, and the authors are appreciated for their efforts to generate these large-scale datasets and functional validation. The proposed “prime, specify and transit” model is also an interesting concept. I do have a few minor suggestions to help improve the paper.

Q1. The authors should introduce more background or supporting evidence for the XEN-like master factors circuitry (Sox17-Sall4-Gata4-Foxa2). This may not be immediately evident for readers outside of reprogramming field.

Response 1: We appreciate the constructive suggestion. We have added a description of XEN-like master factors in this manuscript, which is showed below and cited in the corresponding positions of the article (page 7, line 1-5).

Response figure 6. Core circuitries in mouse ESC cells and XEN cells. Figure 7B, Lim et al., *Cell Stem Cell*, 2008. doi.org/10.1016/j.stem.2008.08.004.

Q2. The authors need to explain C6FAE when first introduced, and why they chose to investigate C6F but left out AE function in this study.

Response 2: We thank the reviewer for pointing out the inadequate points in our manuscript. It was reported that CHIR99021, 616452 and Forskolin were essential compounds to induce CiPSCs and XEN-like cells, and AM580 plus EPZ004777 improved the efficiency of chemical reprogramming (Figure S24 and S30, Hou et al., *Science*, 2013. DOI: 10.1126/science.1239278. Figure 2 and S2, Zhao et al., 2015. DOI: doi.org/10.1016/j.cell.2015.11.017). In our work, C6FAE was a more efficient condition than C6F (response figure 7). However, when investigating the roles of small molecules, we mainly studied the role of core small molecules because AM580 and EPZ004777 were only a plus to enhance the efficiency of XEN-like cell generation and did not determine the induction direction towards XEN-like cells.

Response figure 7. The number of XEN colonies induced by 6FAE, CFAE, C6AE, C6F and C6FAE at day 12.

Q3. The authors stated “we found that cells were quite close to each other during the earlier days and then gradually separated, at last dividing into two branches” - “the proceeding branch” (left branch) and “the trapped branch” (right branch). To me, most d20 cells are in the proceeding branch. Does it mean that d20 cells in the trapped branch largely died? Otherwise this may not be an accurate statement. I also suggest to perform

a differential analysis between pseudo-time and the real-time of the RNA-seq data to better explain the trajectory.

Response 3: We thank the reviewer for pointing out this important aspect. In fact, there were a lot of cells died from day 12 to day 20 after passaging. Cells were passaged on day 12, and after that, senescent fibroblasts could not proliferate, but XEN cells expanded a lot. So, cells on day 20 were mostly XEN cells and most cells on day 20 were in the proceeding branch.

We also performed pseudo-time analysis of single-cell RNA-seq data. Similar with PCA analysis, we also found that cells were divided into 2 branches at the late phase of reprogramming. Interestingly, most d20 cells were still in one branch, but not in both branches. This result further validated the correctness of our PCA analysis and trajectory analysis and was added in the revised version of our manuscript (Extended Data Fig. 1F).

Response figure 8. Scatter plot of the pseudo-time trajectory of individual cells from day 0 to day 20.

Q4. The single-cell RNA-seq data require quality controls. The author may comment on the comparison with existing datasets along the chemical reprogramming process, for example, Zhao et al. (Cell Stem Cell, 2018).

Response 4: We appreciate the reviewer for pointing out the inadequate points. We have compared our single-cell RNA-seq data with existing dataset (Zhao et al., *Cell Stem Cell*, 2018. <http://doi.org/10.1016/j.stem.2018.05.025>), and found that we detected more UMIs and genes than the dataset of Zhao et al. The expression of XEN and fibroblast master genes in various periods were also comparable. Importantly, our MEF cells and d20 cells merged perfectly with Zhao's MEF cells and XEN cells, respectively (response figure 9). These data further verify the reliability of our single-cell RNA-seq data and the correctness of our analysis. We added this comparison data in the revised version of our manuscript and also discussed more about this (page 6, line 2-7, Extended Data Fig. 1A-D).

Response figure 9. Quality control of single-cell RNA-seq data in this study. A-B. Number of detected UMIs (A) and genes (B) comparing with data in Zhao et al. *Cell Stem Cell*. 2018. C-D. Expression level

of MEF (C) and XEN (D) master genes comparing with data in Zhao *et al. Cell Stem Cell*. E. t-SNE of gene expression integrated from data in this study and in Zhao *et al. Cell Stem Cell*. 2018.

Q5. Can the authors perform Sox17 ChIP-seq (if a good antibody is available) or luciferase assay to see whether Sox17 directly regulates the transcription of these downstream factors?

Response 5: We thank for the helpful suggestion. Niakan et al have already found that Sox17 directly bound to the promoter regions of other XEN master genes, including *Sall4* and *Gata4* in XEN cell line by ChIP-seq (response figure 10) (Niakan et al., *Genes Dev*, 2010. DOI: 0.1101/gad.1833510). This meant that Sox17 was able to directly regulate the transcription of these downstream factors. Although we have not determined the regulation of Sox17 on other master transcription factors in the context of chemical reprogramming, we hope the reviewer agree with us that it could be a minor point in our manuscript, which mainly intended to show the stepwise manner of master TF activation and the roles of core chemicals on that.

Response figure 10. Sox17 binds to target genes required for extraembryonic differentiation. Figure 4, Niakan et al., *Genes Dev*, 2010. DOI: 0.1101/gad.1833510.

Q6. Can the authors discuss how likely this “prime-specify-transit” model may be applicable to other chemically induced reprogramming and reprogramming in general?

Response 6: We would like to thank the reviewer's comments for improving our manuscript. We found that the chemical reprogramming progress from fibroblast to neural stem cell could also adopt the “prime-specify-transit” model (Zhang et al., *Cell Stem Cell*, 2016. DOI: 10.1016/j.stem.2016.03.020). According to the RNA-seq data, in the beginning, neural stem cell master gene Sox2 was activated by small molecules, which might prime the cell fate from fibroblast to neural stem cell. After that, 32 neural stem cell-enriched genes, such as *Ascl1*, *Hes5* and *Oligo2*, were activated. In this process, neural stem cell core regulators network was built up and indeed it was a specification process. In the last, fibroblast core genes were erased and cell fate of neural stem cell were built up.

Besides, we found that the “prime-specify-transit” model also existed in the chemical reprogramming process from fibroblast to photoreceptor-like cells (CiPCs) (Mahato et al., *Nature*, 2020. DOI: 10.1038/s41586-020-2201-4). The reprogramming process from fibroblast to CiPCs was primed by the activation of photoreceptor-specifying transcription factors—such as ROR β , ASCL1, PIAS3 at the initiate stage. Then a lot of photoreceptor-specifying genes were activated and the cell fate specification process was completed. It was not until the last stage of reprogramming that the fibroblast cell fate was erased, which was indeed the transition process. Therefore, we believe the “prime-specify-transit” model can be extended to other chemical reprogramming systems.

Importantly, inspired by this model, we induced neuron-like cells, skeletal muscle cells and hepatocyte-like cells from mouse fibroblasts with pure chemicals by fine-tuning small molecule combinations of specification and transition process (unpublished related manuscript). It could be anticipated that more functional cell types could be generated according to this principle, even from human cells in the future for regenerative medicine. We added more discussion on this point in the revised manuscript.

Response figure 11. The “prime-specify-transit” model in the chemical induced neural stem cell reprogramming. Data was adapted from Zhang et al., *Cell Stem Cell*, 2016. DOI: 10.1016/j.stem.2016.03.020

Response figure 12. The “prime-specify-transit” model in the chemical photoreceptor-like cells (CiPCs). Data was adapted from Mahato et al., *Nature*, 2020. DOI: 10.1038/s41586-020-2201-4.

Minor

comments:

Q7. Fig. 1A requires more detailed explanation. What is the difference between the “de-differentiate and re-differentiate model” and “prime, specify and transit model”?

Response 7: We thank the reviewer for this helpful suggestion. The "prime, specify and transit model" is a hierarchical process of obtaining cell fate, and cell fate transition occurs at the end of reprogramming. However, the "de-differentiate and re-differentiate model" involves multiple cell fate changes, and cell fate transition occurs at both the beginning and the end of reprogramming. Compared with “prime, specify and transit model”, “de-differentiate and re-differentiate model” had intermediate cell types with more naive cell state that has the development potential to differentiate into both the initial and product cell types.

Q8. Fig. 2A, D are a bit too small to read. Quantitative analysis should be provided for all IF results.

Response 8: We are sorry for this mistake. Fig. 2A and Fig. 2D were modified according to the suggestion. We had also added quantitative analysis of IF results in our figures.

Q9. The authors stated that “endogenous expression of Sox17 is significantly activated by chemical cocktails in the first 4 days”. This is also clear in Fig. S1E. However, this is not obvious in Fig. S1D. Since both figures are presumably from the same datasets, this needs to be clarified.

Response 9: We are sorry for the confusion of the reviewer. We thought this confusion was caused by the way the data was presented. Gene expression presented in Fig. S1D was normalized by their own maximum levels. To eliminate this confusion, we directly showed the mean value of $\log_2(\text{TPM}+1)$ of each gene within each sample in the revised version, and *Sox17* was found significantly activated (response figure 13).

Response figure 13. Expression of XEN genes at the indicated time points. Color key: $\text{Log}_2(\text{TPM}+1)$.

Q10. It would be nice if the authors can analyze the correlation of the Sox17 expression with Sall4, Gata4, and Foxa2 in the single cell level from their scRNA-seq.

Response 10: We thank the reviewer for the kind remind. We analyzed the correlation of among *Sox17* and other XEN master genes in the single cell level. We found that the level of *Sox17* was highly positive correlated with the expression of *Sall4*, *Gata4* and *Foxa2* in the proceeding branch, but not in the trapped branch (response figure 14). This result also suggested that XEN core regulator network was built up in the proceeding branch.

Response figure 14. The correlation of XEN and MEF genes in the single cell level.

Q11. This might be out of the scope of the paper, but in addition to BMP, can the authors discuss are there any potential new potential master regulators that are upstream of Sox17? This may shed lights on even earlier regulators.

Response 11: We thank the reviewer for the kind suggestion. Indeed, we have previously studied some upstream regulators of *Sox17*. In addition to BMP, we found that *Tbx3* and *Hes6*, which were activated earlier than *Sox17* by small molecules, could regulate the activation of *Sox17* and the knockdown of *Tbx3* or *Hes6* greatly hampered the activation of *Sox17* (response figure 15). So, we think that there are other upstream regulators of *Sox17*, in addition to BMP. We agree with the reviewer that it could be out of the scope of this study, and will not show the data in the revised manuscript.

Response figure 15. Top: Expression of *Hes6* and *Sox17* at different time points; Relative expression of *Sox17* and *Hes6* induced by C6FAE on day 4 with the knockdown of *Hes6*; Relative expression of *Sox17* and *Hes6* induced by C6FAE on day 4 with the overexpression of *Hes6*; Bottom: Expression of *Tbx3* and *Sox17* at different time points; Relative expression of *Sox17* and *Tbx3* induced by C6FAE on day 4 with the knockdown of *Tbx3*; Relative expression of *Sox17* and *Tbx3* induced by C6FAE on day 4 with the overexpression of *Tbx3*.

Q 12. The authors mentioned several manuscripts in submission in the paper. Unless they are accompanied by this submission (or BioRxiv deposition), I suggest to reduce such discussions of unpublished data which are difficult to judge without seeing the data.

Response 12: We appreciate the reviewer for the helpful suggesting. We have already reduced discussions of those unpublished data.

REVIEWERS' COMMENTS:

Reviewer #1 (Remarks to the Author):

In the revised manuscript, the authors performed experiments and provided reliable data for my major concerns mentioned before. This manuscript can be published on Communication Biology as this work will offer a reasonable and beneficial model for chemical modulation of cell fate transition.

Reviewer #2 (Remarks to the Author):

The authors have addressed my previous concerns and the paper is clearly improved over the previous submission. I believe that the manuscript is ready for publication.